# Perceptions of the COVID-19 Pandemic among Women with Infertility: Correlations with Dispositional Optimism

**DOI:** 10.3390/ijerph19052577

**Published:** 2022-02-23

**Authors:** Amanda J. Dillard, Ava E. Weber, Amanda Chassee, Mili Thakur

**Affiliations:** 1Department of Psychology, Grand Valley State University, Allendale, MI 49401, USA; weberav@mail.gvsu.edu; 2Reproductive Genomics Program, The Fertility Center, Grand Rapids, MI 49525, USA; achassee@mrivf.com (A.C.); milithakur@gmail.com (M.T.); 3Department of Obstetrics, Gynecology and Reproductive Biology, College of Human Medicine, Michigan State University, Grand Rapids, MI 49503, USA; 4Department of Obstetrics, Gynecology and Women’s Health, Spectrum Health Medical Group, Grand Rapids, MI 49503, USA

**Keywords:** fertility problems, dispositional optimism, COVID-19 pandemic, fertility treatment, stress and depressive symptoms

## Abstract

People who are more optimistic may experience better psychological health during stressful times. The present study examined the perceptions and emotions surrounding the COVID-19 pandemic among American women who were experiencing fertility problems. We tested if dispositional optimism in these women was associated with less negative perceptions and emotions. We conducted a cross-sectional survey of patients from a single private infertility and reproductive clinic in an urban area in the Midwest, United States. Women, age 18 or older, primarily White and educated, who presented for an appointment to the clinic were invited to participate in an email-based survey. Respondents (*N* = 304) reported their perceived impact of the COVID-19 pandemic on fertility treatment, emotions associated with this impact, and perceived stress and depressive symptoms. They also completed measures of dispositional optimism and expectations for a future pregnancy. Findings indicated that women perceived an overall negative impact of the pandemic on their treatment plans, which was associated with more negative emotions, lower expectations of future pregnancy, and greater stress and depressive symptoms during the pandemic. However, further correlational analyses revealed that being higher in trait optimism was associated with perceiving a less negative impact of the pandemic, experiencing fewer negative emotions, and less overall stress and depressive symptoms. Although women with fertility problems have perceived the pandemic as negative and disruptive, those who are higher in optimism may be less affected.

## 1. Introduction

The coronavirus disease 2019 (COVID-19) pandemic has increased distress among many people [1], and it may be particularly distressing among groups who already experience major stressors in their lives. Women with fertility problems may represent one of these vulnerable groups, as research suggests that they often report high levels of distress about infertility [2,3]. When these women have access to treatment, they experience better psychological health, and this is true even if treatment fails [4]. Beginning in March 2020, fertility clinics across the United States had to close, leaving many women to have to delay long-planned, time-sensitive fertility treatments [5,6]. Because of this disruption to treatment, women with fertility problems may be experiencing the pandemic as particularly distressing. The present study was conducted to understand these women’s perceptions of the pandemic’s impact, their emotions associated with this impact, and whether trait optimism related to their perceptions and emotions. Examining these constructs in this unique group of patients and during a real-world stressor represents a novel contribution to the literature. The findings may further our understanding of the role of trait optimism in helping women with fertility problems amidst other stressful life events.

Research suggests that couples who deal with infertility experience much distress [2,3]. In one review of the literature, researchers found that up to 60% of individuals who experience fertility problems report symptoms of anxiety and depression, and these levels are higher than individuals who do not experience problems [3,7]. Importantly, this distress can be experienced by both men and women and may relate to factors such as low perceptions of control, feelings of guilt or shame, and coping strategies [2,3,8]. Different types of psychological interventions can help to reduce this distress [3] and having access to fertility treatments can also help [4,9]. Unfortunately, recent research suggests that the COVID-19 pandemic may be exacerbating the distress in individuals with fertility problems [10]. Many women with fertility problems report that the pandemic is another key stressor in addition to the major stressor of infertility [11]. Although being in a pandemic may be stressful for many people, the distress may be compounded for women with infertility because it has forced many of them to have to postpone their treatment [10].

Although the COVID-19 pandemic may have increased distress in women with fertility problems, psychosocial variables related to the ways they think and cope may potentially lessen its impact [12]. The personality trait of optimism, for example, may reduce women’s negative perceptions and emotions during the pandemic. Trait optimism is the tendency to hold general positive expectations for one’s life outcomes [13,14]. Decades of research on this trait has revealed that more optimistic people are happier and healthier. For example, higher optimism is correlated with higher self-esteem, more positive affect, and lower anxiety and neuroticism [14,15,16]. More optimistic individuals also experience better physical health outcomes, from daily physical symptoms and risk of various diseases to recovery from medical procedures [13,17,18,19].

Research has also found that being an optimistic person may pay special dividends during stressful times. For example, optimism can reduce distress when dealing with a disease such as breast cancer [20] or when experiencing novel, potentially stressful milestones in one’s life [21]. Trait optimism has also been found to moderate stress-related immunity [22]. For example, optimism is associated with a healthier cellular profile following short-term and long-term stressors.

In the area of pregnancy and fertility, trait optimism has similarly been found to be helpful when experiencing stressful feelings (e.g., [23]). In women dealing with fertility problems, at least two reviews of the literature have highlighted trait optimism as an important factor that can reduce high distress (e.g., [24,25]). Higher optimism has also been associated with less anxiety when women with fertility problems undergo treatment that ultimately fails [4]. One reason for this may be because optimism may allow people to maintain positive expectations for the future even when experiencing a currently stressful situation. For example, in women with fertility problems, although treatment may have failed, more optimistic women may believe they will have higher chances the next time around.

### 1.1. Present Research

In the present research, we examined how women with fertility problems perceived the COVID-19 pandemic. At the time of data collection (July 2020), the pandemic had been occurring in the U.S. for nearly 8 months. Over this course of time, women from across the country had to postpone important fertility treatments they had long been planning [5,6]. This disruption to treatment may have led to considerable distress in these women [10]. In this study, we examined questions such as how impactful do women perceive the pandemic to be on their fertility treatment plans, and to what extent have they experienced various negative emotions over this impact? We also assessed women’s overall stress and depressive symptoms. Given that trait optimism has previously been connected to lower distress in women with fertility problems (e.g., [24,25]), we tested whether trait optimism in these women is associated with less negative perceptions and better psychological health. Specifically, we tested the questions, are women who are higher in trait optimism perceiving the pandemic less negatively and less distressed about its impact? In addition, do more optimistic women have more positive expectations for a future pregnancy? Indeed, this latter idea may be one reason that more optimistic women are less distressed; their optimism allows them to maintain positive expectations about a future pregnancy outcome even in the face of the pandemic.

### 1.2. Sample Size Determination and Data Availability

According to a power analysis using G*Power for correlations using an effect size estimate of *r* = 0.15 and a statistical power of 80%, we would require a sample size of 300 or more respondents. We sought to collect at least that number of participants in the study given time and resource constraints. The data examined in this study is included at an Open Science Framework website in SPSS format: https://osf.io/a9e3w/ (accessed on 1 February 2022).

## 2. Method

### 2.1. Participants

Approximately 648 patients visited the survey website and read the consent form. Of these patients, 241 failed to finish the survey or answer demographic questions, including whether they had been diagnosed with COVID-19 or whether they were currently pregnant. Three respondents reported they had previously had COVID-19. Thus, we did not include these respondents in analyses. For the purpose of this study, we were only interested in women who answered “No” to the question, “Are you currently pregnant” rather than “Yes” (*N* = 63) or “Unsure” (*N* = 37). This left a total sample of 304 women.

Of the 304 respondents for which there was complete (or mostly complete) data, the majority reported they were White (90%; 1% of whom were of Hispanic ethnicity), with some reporting Black or African-American (3%) and Asian (3%). Their average age was 34 years (SD = 4.8), and they were highly educated, with 84% reporting at least a college degree. None of the respondents had been diagnosed with COVID-19. The average length of infertility among respondents was a little over 3.5 years (*M* = 44 months). Nearly 60% of them reported that the major challenge to fertility was “unexplained”, with the second most common category being “age-related factors”. Approximately 70% (*N* = 213) of the participants reported that they had restarted treatment since the clinic had opened.

### 2.2. Procedure

The study was a cross-sectional assessor-blinded study based on patient survey responses submitted anonymously. Data collection occurred over an approximately 1-month period, beginning 23 July 2020. Email invitations were sent to female patients at a single private infertility and reproductive endocrinology clinic in a large urban area in the Midwest, U.S. All patients were older than the age of 18 and had a scheduled appointment between 1 January 2020 and 25 July 2020. The email invited patients to participate in an anonymous online survey titled, “Impact of Novel Coronavirus (COVID-19) Pandemic: Perceptions of Women Receiving Fertility Treatment.” Potential participants were told, “We would like to gain insight into our patients’ feelings, beliefs, and experiences during the novel coronavirus pandemic.” They were asked if they would take a 15-min short survey. We included a link to the study website (maintained by the university) that potential participants could click on if they were interested. Upon going to the link, potential participants could read a consent form. If they agreed to participate, they began the survey. The survey assessed the perceived impact of the pandemic on fertility treatment, emotional response to this impact, stress and depressive symptoms, dispositional optimism, and expectations for pregnancy in the future. Multiple items were used to assess each measure, and composites were created by taking either the sum (following scoring instructions for trait optimism and perceived stress) or an average of items (remaining measures). Participants did not receive any compensation for participating. The study was approved by the University (GVSU) Institutional Review Board.

Note that although the fertility practice from which these patients were recruited did not completely close, all in vitro fertilization procedures (including oocytes retrievals and embryo transfers) that had been scheduled between 27 March and 1 May were deferred. During this time, physicians at the practice continued to see new and established patients via televisits as well as provided ultrasounds for pregnant women.

### 2.3. Measures

*Perceived impact on fertility treatment.* To assess women’s perceptions regarding the degree to which the pandemic has impacted their fertility treatment plans, we developed six items. One item asked, “To what extent has the COVID-19 pandemic negatively influenced your fertility treatment plans?” Participants could respond on a 7-point scale from “Not at all” to “Very significantly.” The five other items were statements that participants could indicate agreement with also on a 7-point scale anchored by “Strongly disagree” to “Strongly agree.” Examples of these statements are, “The pandemic has made me lose so much important time related to my treatment,” “The pandemic has severely impacted my treatment plans,” and “The pandemic has made me feel powerless over my fertility treatment.” We averaged the six items to create a composite (alpha reliability, 𝛼 = 0.88).

*Emotions related to impact on fertility treatment.* Five items were developed to assess the emotional response to treatment impact. Specifically, participants answered questions such as “How worried have you been about your fertility treatment being delayed?” Four other similar questions asked about how hopeless, sad, angry, and disappointed participants were feeling. All questions were on a 9-point scale from “Not at all” to “Extremely.” The items were averaged to create a composite (𝛼 = 0.93).

*Perceived stress.* The Perceived Stress Scale (PSS; [26]) was used to assess respondents’ recent distress during the pandemic. The PSS includes 10 items that ask individuals to report the extent to which their lives have been unpredictable and overloaded. The items are general rather than focused on specific events or experiences. Participants were instructed to “Please indicate how often you have felt or thought a certain way” during the last two months. Examples of questions include “How often have you felt nervous or stressed?” “How often have you felt that you were unable to control the important things in your life?” and “How often have you been upset because of something that happened unexpectedly?” Participants indicated the frequency they felt this way using a scale from “Never (0)” to “Very often (4)”. Four items needed to be reverse-scored, and to create a composite, we summed across the 10 items (𝛼 = 0.77).

*Depressive symptoms.* To assess respondents’ recent depressive symptoms, five questions were adapted from the health status and quality of life measure SF-36 [27]. The items were “During the past four weeks, how much of the time have you felt so down in the dumps that nothing could cheer you up?” “How much of the time have you been a happy person?” “How much of the time have you felt downhearted and blue?” “How much of the time did you have a lot of energy?” and “How much of the time did you feel worn out?” All questions were on a 6-point scale from “All of the time” (1) to “None of the time” (6). The five items were averaged to create a composite of depressive symptoms (𝛼 = 0.86).

*Dispositional optimism.* Optimism was assessed with the Life Orientations Test (LOT-R; [14]). The LOT-R is the most widely used and accepted scale of dispositional/trait optimism. It consists of 10 items, but four items are fillers. Participants indicated their agreement with statements such as “In uncertain times, I usually expect the best,” “I’m always optimistic about my future,” and “I rarely count on good things happening to me.” Participants indicated their agreement on a 0 to 4 scale from “I disagree a lot” to “I agree a lot.” Three items were reverse-coded so that higher scores represented higher optimism. All six items were combined into a summary composite (𝛼 = 0.86).

*Expectations for pregnancy in the future.* To assess respondents’ beliefs about their likelihood of becoming pregnant in the future, two questions were used. Respondents answered the question, “How likely is it that you will get pregnant at some point in the future?” They could respond on a 7-point scale from “No chance” to “Certain to happen.” A second question assessed likelihood in numerical terms: “Please estimate your likelihood of getting pregnant at some point in your life, where 0% means no chance, and 100% means guaranteed.” Respondents could enter any number between 0 and 100. Because these expectations used different scales, we examined them separately rather than combined them.

### 2.4. Data Analysis

Data were analyzed using the Statistical Package for the Social Sciences (SPSS version 26, IBM, Armonk, NY, USA). To examine the women’s perceptions of the impact of the pandemic on their fertility treatment plans, their associated negative emotions, and overall stress and depressive symptoms due to the pandemic, we computed descriptive statistics (means and standard deviations) of composite scores. Correlation analyses were used to determine the associations among primary variables as well as to test associations between trait optimism and primary variables. Regression analyses were also conducted to control for demographics and confounding associations among variables.

## 3. Results

### 3.1. Descriptives

None of the patient demographics were significantly correlated with primary variables. In other words, demographics such as age, race, or education were not related to perceived impact on fertility treatment, the emotions associated with the impact, or trait optimism. Fertility variables such as length of time of infertility were also not correlated with primary variables.

Table 1 shows the means and SDs of primary variables collapsed across demographics. The means for both perceived impact on fertility treatment and emotions associated with this impact were higher than the midpoint of the scales, suggesting respondents on average were reporting moderate levels of perceived impact and negative emotion. Perceived stress and depressive symptoms were also higher than midpoints for the scales and, notably, the perceived stress mean among this group was higher than what has been observed for normal populations (e.g., 19.90 for the present sample vs. ~13.5 for females and this age range; [28]).

### 3.2. Primary Analyses

Table 2 shows correlations among the primary variables. Several significant associations emerged. First, perceived impact on fertility treatment was significantly associated with a negative emotional response. In other words, as women perceived the pandemic to have a more negative impact on their treatment plans, they reported experiencing more negative emotions (e.g., anger, sadness, etc.) about treatment. Both perceived impact and this emotional response were significantly associated with women’s reports of recent stress and depressive symptoms. The associations were in the predicted directions such that as women perceived the pandemic as having a more negative impact on their treatment and they experienced more negative emotion related to this impact, they also reported more recent stress and depressive symptoms.

### 3.3. Associations with Optimism

We next examined the correlations with trait optimism, which are also shown in Table 2. As predicted, trait optimism was significantly negatively associated with perceived impact on fertility treatment. Specifically, as women scored higher on trait optimism, they perceived the pandemic as having less of a negative impact on their fertility treatment. Being higher in optimism was also associated with reportedly experiencing less negative emotion about the pandemic’s impact on treatment. Finally, higher optimism was also significantly associated with reporting lower recent stress and depressive symptoms.

Although the correlational analyses showed that trait optimism was significantly related to primary variables, given the covariance among variables, we conducted hierarchical regressions to examine independent associations between trait optimism and each primary variable (perceived impact, emotional response, stress, depressive symptoms). In each regression, we examined the association between optimism and a given primary variable while entering demographic variables (age, race, education) in Step 1 and entering the remaining primary variables in Step 2. With the exception of perceived impact, analyses revealed that all associations were significant. The associations showed that as trait optimism was higher, there was less negative emotion about the pandemic’s impact, B = −0.38, SE = 0.17, *b* = −0.16, *t*(288) = −2.27, *p* = 0.024, as well as less stress, B = −0.24, SE = 0.08, *b* = −0.24, *t*(288) = −3.21, *p* = 0.002, and depressive symptoms, B = −1.72, SE = 0.47, *b* = −0.28, *t*(288) = −3.70, *p* < 0.001.

Finally, we analyzed women’s expectations for becoming pregnant in the future. Trait optimism may be associated with more positive expectations for pregnancy in the future, and this may be one reason for better psychological health during the pandemic. Table 2 shows these associations. Both pregnancy expectation items were positively associated with trait optimism: As women scored higher in trait optimism, they had more positive expectations for a future pregnancy.

## 4. Discussion

In the present study, we examined how women with fertility problems were experiencing the COVID-19 pandemic. Overall, the findings showed that these women perceived a highly negative impact of the pandemic on their fertility treatment plans, and correspondingly they reported experiencing high rates of various negative emotions. Significantly, trait optimism was associated with these perceptions and emotions, suggesting this trait may be a protective factor. This study is one of the first to examine how this clinical population is faring with the COVID-19 pandemic. While our findings are consistent with other recent research that has found a negative impact of the pandemic on these women’s psychological health [10,14,29], our data suggest that optimism is a factor that likely relates to this impact.

In the domain of pregnancy and fertility, trait optimism has been found to be a beneficial factor. For example, the trait has been associated with more positive physical health outcomes such as lower risk of endometriosis [30] and lower risk of pre-term birth [31,32]. In women who are undergoing fertility treatment, higher trait optimism has been associated with better psychological health and more positive physical responses to treatment (e.g., [4,33]). In the present study, we found that trait optimism was associated with better psychological health in women dealing with fertility problems and a pandemic. More optimistic women were better off both in terms of their specific emotions about the pandemic’s impact and their general stress and depression due to the pandemic. These associations are consistent with the prediction that trait optimism may protect individuals during times of high stress.

One of the biggest questions stemming from our findings is why was trait optimism associated with better responses to the pandemic? One reason may relate to self-efficacy or confidence in one’s ability to bring about a desired outcome [34]. Self-efficacy has been connected to optimism [35], including recently in women who have fertility problems [36]. These constructs may work together, such that people who are higher in trait optimism may have more confidence in their ability to bring about desired outcomes. In the case of our participants, even though the COVID-19 pandemic may have disrupted their treatment plans, those who were more optimistic may have been better able to imagine achieving their future outcome of pregnancy. Their optimism was associated with seeing this desired outcome as still completely achievable. Indeed, findings showed that the higher the women’s optimism, the greater their expectations of becoming pregnant in the future. Thus, in the face of significant stress, trait optimism was associated with maintaining positive expectations about the future, including the desired outcome of becoming pregnant.

Another reason that trait optimism tends to be associated with psychological health relates to coping. Much evidence has accumulated to suggest that people who are higher in trait optimism cope more effectively with life stressors. One good example of this is that people who are higher in trait optimism have been found to be more likely to use problem-focused rather than avoidant coping when facing real-life health problems (for reviews, see [37,38]). This research also suggests that when feeling stressed, optimists are less likely to use unhealthy behaviors, such as drinking alcohol, to cope. To date, there have been very few studies to examine how perceptions of the pandemic (including closure of clinics) are related to coping in women with fertility problems. In one study, researchers found that while some patients reported coping moderately well, nearly 12% were coping poorly and reporting intense negative feelings [10]. The participants in our study who reported higher optimism may be less likely to be in the latter group, coping more effectively with the stress of infertility and the pandemic in general.

Although the present study showed that trait optimism was associated with several indices of psychological health, one question for future research is whether this optimism may confer positive behavioral or physical health outcomes down the line. For example, after the pandemic is over, might women who are more trait optimistic be quicker to restart treatment or have a better response to treatment, such as becoming pregnant? Moreover, optimism likely interacts with other traits to influence such outcomes. In one study, higher dispositional optimism along with lower trait neuroticism were associated with a more positive biological response to fertility treatment [33]. Trait optimism, as well as other personality traits, might be assessed in the future to test how their interactions influence these outcomes.

If more optimistic women are faring better during this pandemic, as our data and recent others might suggest, how could clinicians cultivate more optimism in their less-than-optimistic patients? One strategy that could possibly increase at least temporary optimism is the use of the Best Possible Self (BPS) exercise. In the BPS exercise, individuals are asked to write, for a specified amount of time, about an ideal future life in which everything turns out for the best and all of their goals have been achieved [39]. King found that this strategy increased positive affect, physical health, and general wellbeing (also see [40]). These positive visualizations could increase optimism and self-efficacy [40,41]. Given these virtual times, including patient-doctor interactions, one important caveat is that psychological interventions to increase optimism may be significantly more impactful when they are conducted in-person rather than online [42].

This study was not without limitations. First, this was a non-probabilistic sample, and many participants were White and of higher socioeconomic status (SES), evidenced by their high education level. Research has connected SES with dispositional optimism in women with fertility problems [36]. It is unclear whether dispositional optimism would show these associations in a sample of lower SES fertility patients. Second, while we offer an explanation that higher optimism may be related to doing better psychologically because of higher self-efficacy, we did not measure self-efficacy. Instead, we measured expectations for a future pregnancy, which may be an indirect proxy of the construct. A third limitation is that our measure of depressive symptoms was not a typical depression measure, but part of a quality-of-life assessment. However, other research has used the SF-36 to assess depressive symptoms during the COVID-19 pandemic [43]. Two more limitations are general in nature. First, all measures in the present study were self-reported and are vulnerable to social desirability bias and recall errors. Finally, these data were correlational and, thus, causal attributions cannot be made. In other words, it is unclear whether optimism led to less negative perceptions and emotions or these perceptions and emotions led to greater optimism. Future studies that test the effects of optimism interventions on psychological health in this patient population could offer insight into these causal links.

## 5. Conclusions

In the present research, we found that although women perceived an overall negative impact of the COVID-19 pandemic on their fertility treatment and experienced various negative emotions as a result, women who were higher in trait optimism had fewer negative perceptions and emotions. Although trait optimism has previously been associated with less distress in women with fertility problems (e.g., [25]), this is the first study to show that optimism is associated with less distress in the crux of an additional novel stressor, including one that has threatened access to treatment. Future research should examine questions such as whether trait optimism is associated with better responses to treatment following the pandemic or how clinicians can cultivate optimism with their patients during novel stressful times.

## Figures and Tables

**Table 1 ijerph-19-02577-t001:** Means and standard deviations for primary variables.

	*Ms*	*SDs*
Perceived impact on fertility treatment	5.01	1.40
Emotions related to impact on fertility treatment	5.79	2.28
Dispositional optimism	14.08	5.31
Expectations for pregnancy in the future	4.33	1.55
Expectations for pregnancy in the future (numerical)	57.48	28.09
Perceived stress	19.90	5.25
Depressive symptoms	2.91	0.85

*Note.* Perceived impact on fertility treatment score was on a 7-point scale, with higher numbers representing greater negative impact. Emotions related to impact score was on a 9-point scale, with higher numbers representing more frequent negative emotion. Dispositional optimism scores ranged from 0 to 24, with higher numbers representing more optimism. Expectations for pregnancy in the future was on a 7-point scale, with higher numbers representing greater likelihood, and numerical expectations were on a 100-point scale. Perceived stress scores ranged from 4 to 32 (possible highest score was 40), with higher numbers representing more stress. Depressive symptoms score was on a 6-point scale, with higher numbers representing more symptoms.

**Table 2 ijerph-19-02577-t002:** Correlations among primary variables.

	1	2	3	4	5	6
1. Perceived impact on fertility treatment	-					
2. Emotions related to impact on fertility treatment	0.71 **	-				
3. Dispositional optimism	−0.17 **	−0.26 **	-			
4. Expectations for pregnancy in the future	−0.14 *	−0.10	0.25 **	-		
5. Expectations for pregnancy in the future (numerical)	−0.17 **	−0.09	0.15 *	0.76 **	-	
6. Perceived stress	0.16 **	0.24 **	−0.46 **	−0.10	−0.05	-
7. Depressive symptoms	0.17 **	0.22 **	−0.47 **	−0.15 *	−0.18 **	0.73 **

Note. ** *p* < 0.01; * *p* < 0.05.

## Data Availability

The data that support the findings of this study are publicly available on the Open Science Framework at https://osf.io/a9e3w/ (accessed on 1 February 2022).

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
