# Peer review of "Perceptions of the COVID-19 Pandemic among Women with Infertility: Correlations with Dispositional Optimism"

_ijerph, 2022, doi:10.3390/ijerph19052577_

Round 1

Reviewer 1 Report

  1. There is only a brief history provided in the introduction, and it is unclear what new data the study adds to the existing literature. A more objective introduction that offers more than a brief history is suggested, with the request that the novelty of this study is emphasized more.
  2. In the present study, we carefully examined how women with fertility problems perceive the COVID-19 pandemic. Specifically, how impactful do they perceive the pandemic on their fertility treatment plans, and to what extent have they experienced various negative emotions over this impact? We also assess women’s overall stress and depressive symptoms related to the pandemic. Finally, we test whether trait optimism in these women is associated with less negative perceptions and better psychological health. We test questions such as, are the women who are higher in trait optimism perceiving the pandemic less negatively, and are they less distressed about the pandemic and treatment? Also, do more optimistic women have more positive expectations for a future pregnancy? Indeed, this latter idea may be one reason that women who are more optimistic more optimistic womens are less distressed; their optimism allows them to maintain positive expectations about a future pregnancy outcome even in the face of the pandemic. Please make the suggested changes in italics and bold, and set the purpose of this research more specifically.
  3. Reading carefully that in the present study, the authors started with 648 patients who visited the survey website and read the consent form. Of these patients, 244 failed to answer questions about being currently pregnant and ever having been diagnosed with COVID-19 and 304 respondents for which there was complete data, most reported they were White (90%; 1% of whom were of Hispanic ethnicity), with some reporting Black or African-American (3%) and Asian (3%). Their average age was 34 years (SD = 4.8), and they were highly educated with 84% reporting at least a college degree. Nowhere in the paper did I find the sample size calculation or statistical power. Of the two, I recommend introducing the calculation of statistical power for all three stages in the form:
  • Statistical Power: 80% or 0.80. For a given experiment with these defaults, we may be interested in estimating a suitable sample size. That is, how many observations are required from each sample in order to at least detect an effect of 0.80 with an 80% chance of detecting the effect if it is true (20% of a Type II error) and a 5% chance of detecting an effect if there is no such effect (Type I error).
  1. I propose the following sentence: Although we did not measure self-efficacy directly, we did not measure the women’s expectations of a future pregnancy should be reformulated and introduced separately to the limitations of the study together with This study was not without limitations. First, many participants were primarily White and of higher socio-economic status (SES), as evidenced by their high education level. Research has connected SES with dispositional optimism in women with fertility problems [32]. It is unclear if dispositional optimism would show these associations in a sample of lower SES fertility patients. A second limitation is that these data were correlational, and thus causal attributions cannot be made. In other words, it is unclear if optimism led to less negative perceptions of the pandemic and less negative emotions, or if these perceptions and emotions led to greater optimism. However, given that dispositional optimism is viewed as trait-like and stable over time, research would suggest the former. Nevertheless, future studies that test the effects of optimism interventions on psychological health in this patient population could offer insight into these causal links. Please also make the suggested changes with bold and strikethrough.
  2. Please also make the suggested changes with bold and strikethrough. In the present study, we examined how women with fertility problems were experiencing the COVID-19 pandemic. Overall, the findings showed that these women perceived a highly negative impact of the pandemic on their fertility treatment plans, and correspondingly they reported experiencing high rates of various negative emotions. Importantly Significantly, trait optimism was associated with these perceptions and emotions, which may suggest suggesting this trait was a protective factor. This study is one of the first to examine how this clinical population is faring with the COVID-19 pandemic. While our findings are consistent with other recent research that has found a negative impact on these women’s psychological health [7, 24, 25], our data suggest that optimism may be a factor that can offset this impact.
  • In the present research, we found that although women perceived an overall negative impact of the COVID-19 pandemic on their fertility treatment and experienced various negative emotions, as a result, those women who were higher in trait optimism had fewer negative perceptions and emotions. Although trait optimism has previously been found to be associated with less distress in women with fertility problems [e.g., 20], this is the first study to show that optimism is also protective in the crux of an additional novel stressor, including one that has threatened access to treatment. Future research should examine questions such as whether trait optimism is associated with better responses to treatment following the pandemic or how clinicians can cultivate optimism with their patients during novel stressful times.

I want to mention that for the analysis of the similarity coefficient, I used Plagiarism CheckerX, version 6.0.11. Furthermore, I did not find any significant problem from this point of view.

Reviewer 2 Report

The authors are encouraged to consider the following suggestions.

1. In the Introduction, please describe why only targeted on women but not men. I believe that men with infertility issues are also likely to suffer from psychological distress due to COVID-19 pandemic.

2. The inclusion and exclusion criteria should be explicitly mentioned in the Methods section. Until I read the Results section, I know that the authors have set some inclusion and exclusion criteria. However, this is not mentioned in the Methods section at all.

3. Line 144. Why do the authors describe "alpha reliability" here, while this description is not used for other measures?

4. ll129-137. Using SF-36 items to assess depression symptoms is problematic. Specifically, SF-36 items are designed to assess an overall quality of life. Depression symptoms may be part of the quality of life; however, no prior evidence provides any information regarding how the items on SF-36 represents depression symptoms. Therefore, using some F-36 items here is problematic. The authors should acknowledge this as a limitation.

5. The authors only used correlation analyses to test for their hypothesis. However, simply using correlation analyses cannot control the potential confouding effects. Therefore, the authors are strongly recommended to construct regression models to control the confouder effects.

6. The authors should mention one more limitation that all the measures were self-reports. Therefore, social desirability bias or recall bias may impact on the findings. 

Reviewer 3 Report

The paper defines a study in which we want to explore, through a simple correlational analysis, the extent to which optimism, seen as a dispositional trait, may have had a protective function in a sample of pregnant women in the USA during the COVID-19 period. Although the idea of the study may be nice (but not innovative), the proposed analysis leads to an important reflection on the results. As explained in the limitations, the correlation is based on a two-way association between two variables and does not allow a causal relationship to be inferred. In my opinion, the study is too "simplified" to be published. To state what is said in the title I would expect much more sophisticated and complex analyses, including a probability sample. For this reason, I cannot consider the paper as acceptable in its present form, it should, in my opinion, be thoroughly restructured. Furthermore, I see obvious gaps in the literature part: there is no theoretical reference model, so on what basis do I aim to test whether optimism has protective validity?

1) I recommend that you avoid putting the declarations purpose, method, etc., on the list. A more discursive abstract would be better, making the parts more coherent without defining them in advance. Moreover, in the abstract, there is no mention of where the survey was carried out nor of the socio-demographic characteristics of the sample, nor of how the data was analysed. 
2) The beginning of the abstract, with that "To examine" is not very nice. I suggest starting the abstract in another way, following the classic structure, for example the theoretical-historical context in which the study fits.
3) The introduction seems rather poor. I suggest that the research should be contextualised first. Where was it carried out? What were the characteristics of COVID-19 in that country? A bit of background to define the socio-psychological framework of the environment in which the survey took place would not hurt.
4) Optimism, in itself, is seen as a protective construct in many theories, see for example Bakker and Demerouti's Job Demands-Job Resources model. The fact that there is no theoretical formulation underlying the present study is somewhat disturbing. 
4) In the procedure, I suggest explaining how you created the composite indicator (mean, sum?). It must be made explicit BEFORE the measures. 
5) Table 2. There is no need to put 1's on the long diagonal, it is preferable to put simple dashes "-".
6) Another essential aspect in the limits: the sample is not probabilistic, what kind is it? This clearly has an effect on the generalisation of the results.
7) In case the other reviewers decide to consider the article suitable for future consideration, I strongly suggest to modify the title CLEARLY EXPLAINING that it is a correlational study, without any pretence of generalisation. 

Reviewer 4 Report

Dear authors,

Congratulations for the work, the topic is very original and important in these times.

I would like you to answer the following questions about your article if you are so kind.

  1. How were perceptions of the pandemic examined? Is there a specific questionnaire on perceptions of covid at present or was a questionnaire of your own developed?
  2. Was the survey structured in a self-reporting manner as it was conducted online. Have you considered bias, sample loss in the process? how many subjects were lost? Did all 304 women chosen answer the questionnaires? They should indicate the process with a flow chart.
  3. How was the sample size calculated?
  4. Are there any similar studies on this subject?
  5. How would you do a future treatment to improve dispositional optimism?
  6. Would you continue with the same sample to see if they finally manage to get pregnant and the state of stress and depression decreases? Future studies could consider this option.
  7. Do they distinguish between depressive symptoms and depression as such?

Introduction:

Are there studies on optimism and fertility in general or optimism and effectiveness of fertility treatments? Should be included in the introduction

Methods:

How are the five questions chosen from the SF-36 to measure perceived stress scored or corrected?

Were logistic regressions used in the statistical analysis? How was interference between variables controlled for?

Results:

How was the analysis of the demographic variables done to know whether or not they were associated with the study variables?

Discussion:

What are the coping strategies that were observed in the participants who presented high trait optimism? How can the strategies be seen if the person is not physically seen?

Thank you very much

Round 2

Reviewer 2 Report

The authors have addressed all my prior concerns. However, some minor revisions are needed.

1. Table 2. In order to have consistency, please change the 1 in the first row (i.e., showing the correlation between 1 and 1. Perceived impact fertility treatment) to "-".

2. Correcting the sentence "A third limitation is that our measure of depressive symptoms differed from traditional depression measures. However, other research has used this measure to assess depressive symptoms during the COVID-19 pandemic", to " A third limitation is that our measure of depressive symptoms was a proxy instead of a standardized depression measures. Therefore, the information on depressive symptoms might be measured with bias, although other research has used the SF-36 to assess depressive symptoms during the COVID-19 pandemic." After all, SF-36 is not a measure on depressive symptoms. 

Author Response

The authors have addressed all my prior concerns. However, some minor revisions are needed.

  1. Table 2. In order to have consistency, please change the 1 in the first row (i.e., showing the correlation between 1 and 1. Perceived impact fertility treatment) to "-".
  2. Correcting the sentence "A third limitation is that our measure of depressive symptoms differed from traditional depression measures. However, other research has used this measure to assess depressive symptoms during the COVID-19 pandemic", to " A third limitation is that our measure of depressive symptoms was a proxy instead of a standardized depression measures. Therefore, the information on depressive symptoms might be measured with bias, although other research has used the SF-36 to assess depressive symptoms during the COVID-19 pandemic." After all, SF-36 is not a measure on depressive symptoms. 

Thank you for the suggestions. We have now made these edits.

Reviewer 3 Report

Dear Authors,

thank you for considering my comments. They were based on a constructive point of view, of course. I found your manuscript much improved. I have only a new doubt on the title: should the present title "Perceptions of the COVID-19 pandemic among women with infertility: associations with dispositional optimism" be replaced by another one, as "Perceptions of the COVID-19 pandemic among women with infertility: a correlational study on the role of dispositional optimism"?

Author Response

Dear Authors,

thank you for considering my comments. They were based on a constructive point of view, of course. I found your manuscript much improved. I have only a new doubt on the title: should the present title "Perceptions of the COVID-19 pandemic among women with infertility: associations with dispositional optimism" be replaced by another one, as "Perceptions of the COVID-19 pandemic among women with infertility: a correlational study on the role of dispositional optimism"?

Thank you for the comments. They were very helpful in improving the manuscript. A previous reviewer did not seem to like the ‘role’ terminology. However, we have now revised the title to “Perceptions of the COVID-19 pandemic among women with infertility: correlations with dispositional optimism". We hope that you will find this acceptable.

Reviewer 4 Report

Dear authors, thank you very much for the modifications made.
Best regards

Author Response

Dear authors, thank you very much for the modifications made.
Best regards

Thank you for the comments.